# Associations of Dietary Intake with Cardiovascular Risk in Long-Term “Plant-Based Eaters”: A Secondary Analysis of a Cross-Sectional Study

**DOI:** 10.3390/nu16060796

**Published:** 2024-03-11

**Authors:** Boštjan Jakše, Uroš Godnov, Zlatko Fras, Nataša Fidler Mis

**Affiliations:** 1Independent Researcher, 1000 Ljubljana, Slovenia; 2Department of Computer Science, Faculty of Mathematics, Natural Sciences and Information Technologies, University of Primorska, 6000 Koper, Slovenia; uros.godnov@gmail.com; 3Division of Medicine, Centre for Preventive Cardiology, University Medical Centre Ljubljana, 1000 Ljubljana, Slovenia; zlatko.fras@kclj.si; 4Faculty of Medicine, University of Ljubljana, 1000 Ljubljana, Slovenia; 5Ministry of Health, 1000 Ljubljana, Slovenia; natasa.fidler-mis@gov.si

**Keywords:** plant-based diet, whole food, plant based, active lifestyle, food groups, macronutrients, cardiovascular risk factors, LDL cholesterol

## Abstract

A plant-based diet rich in whole foods and fiber is beneficial for cardiovascular (CV) health. This impact is often linked to specific food groups and their preparation methods, reflecting the overall dietary pattern. However, research on the long-term effects of a carefully designed plant-based diet on adults transitioning from a typical Western lifestyle is limited. Notably, studies on people managing CV risk factors effectively are scarce. As part of a cross-sectional study, we examined 151 individuals committed to a long-term, well-designed plant-based diet and active lifestyle. We investigated how specific food groups and macronutrient intake are related to various CV health markers. In this secondary analysis, our comprehensive approach encompassed several methods: 3-day weighted dietary records, fasting blood lipid and blood pressure measurements, body composition assessments, and evaluations of lifestyle status. We adjusted our analysis for multiple variables, such as age, sex, current body mass index, smoking status, physical activity, and time (years) following the plant-based diet. Our findings revealed several associations between macronutrient intake (per 50 g) and CV risk markers, although these associations were generally weak. Individuals who consumed more whole grains and fruits had lower levels of total, low-density lipoprotein (LDL-C), and high-density lipoprotein (HDL-C) cholesterol. We also found associations between the intake of legumes and nuts/seeds and reduced HDL-C levels. These findings suggested that these food groups might influence the lipid profile, contributing to CV health in a plant-based diet. A greater intake of spices/herbs was associated with lower uric acid levels, while diets rich in plant-based fast food and pasta (made from white flour) were associated with higher uric acid levels. A greater intake of various macronutrients, such as fiber, carbohydrates (from whole-food sources), proteins, and different types of fats (saturated fatty acids [SFAs], monounsaturated fatty acids [MUFAs], and polyunsaturated fatty acids [PUFAs]), was associated with lower levels of total cholesterol, LDL-C (only for carbohydrates), and HDL-C. We found a unique negative correlation between PUFA intake and LDL-C, suggesting that PUFAs might significantly affect LDL-C levels. In contrast, increased fiber, protein and SFA consumption were associated with increased uric acid levels. These findings support the impact of dietary patterns on CV risk factors, highlighting that even small amounts of unhealthy food groups can significantly influence specific CV risk markers, regardless of the overall diet.

## 1. Introduction

Over the past two decades, various dietary patterns have garnered considerable attention. One of these patterns is the exclusive plant-based (vegan) diet, which is characterized by the complete exclusion of all animal products [1]. Once considered a niche dietary choice, plant-based diets, particularly vegetarian diets, have recently surged in popularity due to their reported health benefits [2,3,4,5,6] and contribution to environmental sustainability [7,8,9].

However, there is some confusion about the definition of a “plant-based diet”. This term is used interchangeably with vegan or vegetarian diets in certain studies. However, in other studies, it refers to a diet mainly consisting of plant-based foods. In some cases, it may include animal products in smaller amounts than a typical omnivorous diet recommended by the governmental healthy nutrition guidelines [1,10]. Table 1 describes the basic plant-based dietary patterns.

Cardiovascular diseases (CVDs) are the leading cause of global public health challenges responsible for a substantial burden of mortality and disability globally [16], as well as in Slovenia [17]. The interplay between diet and CV health is well established, making dietary interventions a crucial component of preventative and therapeutic strategies [18,19,20,21,22]. Several dietary patterns, such as plant-based diets, the Mediterranean diet, and the Nordic dietary pattern, have been found to be effective in reducing the risk of CVDs [2,23,24,25,26,27,28]. The common characteristics of these nutritional patterns are the consumption of whole-food groups, reduced or eliminated animal food intake, and increased consumption of plant-based foods. In contrast, Western dietary habits are characterized by excessive energy intake and the overconsumption of added and/or free sugars, added fats, saturated fatty acids (SFAs), dietary cholesterol, and salt, while at the same time lacking sufficient whole foods (e.g., whole grains, legumes, fruits, vegetables, and nuts). This nutritional imbalance contributes to the increasing incidence of obesity, type 2 diabetes, hypertension, hyperlipidemia, and coronary artery disease [29,30,31]. The abundance of fiber, antioxidants, and phytochemicals in plant-based and healthy plant-based diets may contribute to decreased blood pressure, improved lipid profiles, and reduced risk of CVDs [32,33,34,35].

By exploring how different food groups affect CV health, we found that some groups played a pivotal role. Understanding which food groups play a crucial role in CV health is critical in the formulation of effective dietary strategies for the prevention and management of CVDs. Whole grains, legumes, fruits, vegetables, and nuts/seeds consistently show protective effects against CVDs. These foods are packed with fiber, which can help to reduce cholesterol levels, lower blood pressure, and improve blood sugar control [19,36,37,38,39,40,41,42,43]. It is essential to understand that not all plant-based diets are the same. Just as omnivorous diets differ significantly in terms of nutritional sufficiency and health benefits, plant-based diets also fall along a broad spectrum, from very healthy to less nutritious, and consequently, having more or less favorable health benefits [14,44]. An unhealthy plant-based diet might include many highly processed foods, sweetened drinks, and plant-based versions of unhealthy junk foods, resulting in a less healthy dietary pattern [45,46]. Therefore, specific food groups within the plant-based diet, as well as meal preparation methods, require careful consideration. For example, ultra-processed plant-based foods, such as plant-based fast food and sweets, tend to contain high levels of sodium, added sugars, and unhealthy fats. Consuming these foods regularly, and in excessive amounts, can lead to weight gain, insulin resistance, and high blood pressure, increasing the risk of CVDs [47,48,49,50].

In nutrition science, contradictions regarding the associations between various food groups or macronutrients and CVD are common. These disparities can be attributed to several factors, such as differences in study design; sample size; inclusion criteria; methods used to assess dietary intake (which can impact the accuracy of reporting); the duration participants adhere to the diet; quantity, quality, and diversity of the diet; and the influence of lifestyle factors, such as physical activity (PA), smoking, alcohol consumption, and age. Therefore, these factors can impact how diet relates to various CV risk factors. Our previous study compared the nutritional, cardiovascular, and lifestyle status of health-conscious adults following plant-based and omnivorous diets. We observed significant differences in dietary intake (e.g., fiber and SFA intake) and CV risk factor status (e.g., LDL-C levels), although the dietary intake and LDL-C levels in both groups were within the recommended ranges. Using multivariate linear regression analysis, we estimated that the combined effects of fiber, SFA intake, and age accounted for 47% of the variability in LDL-C levels [51].

The present study aimed to investigate the relationships between dietary intake (i.e., specific food groups and macronutrients) and several CV health markers in individuals who adhered to a long-term, well-designed plant-based diet and practiced an active lifestyle. This secondary analysis can provide important contextual insight into which food groups have a distinctive effect on CV risk factors, even when consumed in small amounts. We hypothesize that the consumption of a well-designed plant-based diet is generally associated with favorable CV health profiles. A well-designed plant-based diet is characterized by the consumption of a variety of plant-based food groups daily and weekly. The meals are prepared using healthy cooking methods, such as boiling, steaming, or baking on parchment paper. Therefore, the definition of “well-designed” (also called “healthy”) must meet core dietary principles, such as adequacy, diversity, balance, and moderation, ultimately aiming to satiate and provide health benefits [11,14,15,52,53,54,55].

## 2. Materials and Methods

### 2.1. Study Design and Eligibility

A study was conducted on healthy adults who followed the WFPB lifestyle program. This study utilized various methods, including 3-day weighted dietary records, fasting blood lipids and blood pressure measurements, body composition assessments using a medically approved and calibrated electrical bioimpedance monitor, and evaluations of lifestyle using standardized questionnaires for physical activity (Long International Physical Activity Questionnaire (L-IPAQ), sleep (Pittsburgh Sleep Quality Index (PSQI), and stress (30-question Perceived Stress Questionnaire (PSQ)) [56,57,58]. The methods used are briefly described in Section 2.3, while detailed information about the methodology and the enrolment process can be found in previous publications [59,60,61]. This cross-sectional study was conducted in Slovenia, a European Union member, from June to August 2019. All participants were provided a complete explanation of this study, and written informed consent was obtained from all participants.

The study protocol received approval from the Slovenian Ethical Committee on the Field of Sport (approval document No. 05:2019; the application was submitted on May 16th, 2019, and was approved on 27 May 2019) and the National Medical Ethics Committee of Slovenia (approval document 0120-380/2019/17; the application was submitted on 7 July 2019, and was approved on 20 August 2019). The trial was registered with ClinicalTrials.gov under the number NCT03976479, with registration on June 6th, 2019 (this study was submitted on 1 May 2019, and passed the quality control review by the National Library of Medicine on 4 June 2019).

### 2.2. Subjects

Our final data analysis included 151 adults, consisting of 109 females and 42 males. Study participants adhered to a plant-based diet, specifically the WFPB lifestyle program, for a duration ranging from 0.5 to 10 years (average of four years) [59,60,61]. Before adopting the WFPB lifestyle, these participants followed a Western diet/lifestyle. The primary motive for transitioning to the WFPB lifestyle was to achieve health benefits and manage body mass effectively [61]. The program was designed for primary prevention. Therefore, none of the participants were simultaneously using medications for lipids, blood pressure, or blood sugar control [59].

### 2.3. Participant Characteristics, Nutritional Status, CV Risk Factor, and Lifestyle Status Assessment

Our study assessed participant characteristics, including sociodemographic, economic, and lifestyle status, using the questionnaire developed by the National Institute of Public Health of the Republic of Slovenia [62] and standardized electronic questionnaires [56,57,58,61]. To determine body composition, we employed an 8-electrode bioelectrical impedance body composition monitor (Tanita 780 S MA, Tanita Corporation, Tokyo, Japan) that has been medically approved, calibrated, and validated across several demographic conditions [61,63,64,65]. We utilized the abovementioned body composition analyzer for body mass assessment, while body height was measured using a body height gauge (Kern MPE 250K100HM, Kern and Sohn, Balingen, Germany). The characteristics of mean age, sex, current BMI, smoking status, PA status, and years on a WFPB lifestyle were adjusted for in our analysis when examining our primary objectives.

Nutritional status, which encompasses dietary intake and body composition, was evaluated using 3-day weighted dietary records and the dietary software Open Platform for Clinical Nutrition (OPEN) [66,67], a web-based application developed by the Jozef Stefan Institute [68] and supported by the European Food Information Resource Network (EuroFIR); the European Federation of the Association of Dietitians (EFAD); the European Society for Paediatric Gastroenterology, Hepatology, and Nutrition (ESPGHAN); and the American Dietetic Association (AND) [66]. The participants were given detailed instructions on recording their food intake for three consecutive days. The participants were asked to weigh all consumed foods and beverages using our calibrated electronic kitchen scales and to record the type, amount, and flavor of plant-based meal replacements (PB MRs) and dietary supplements. Semiquantitative recording of standard household measures or a picture booklet was allowed when exact weighing was impossible [69]. We used a recipe simulator to estimate the energy and nutrient contents of commercial or home-prepared foods, which involved the use of labeled ingredients and nutrient contents. Participants were asked to specify the form in which they weighed the foods (raw or cooked) and the form in which they consumed them (e.g., raw, cooked in water, baked in the oven). We used appropriate conversion factors [70] when entering the dietary data into the OPEN. Finally, nutritional intake from dietary supplements plus PB MRs was calculated by Res-Pons, a company that professionally manages a database with all dietary supplements and medicinal products available in the Slovenian market [71]. For this study, the average intake of calories and nutrients is presented as the sum of intakes from food, dietary supplements, and PB MRs.

As part of this study, 10–15 mL of blood was taken from participants after a 10–12 h fast. Lipid levels and other biochemical parameters were assessed using conventional laboratory procedures at national medical biochemical facilities employing uniform analytical methodologies. Total cholesterol (total C), HDL-C, and triglyceride plasma levels were directly measured, while LDL-C levels were determined using the Friedewald formula. Blood pressure was assessed using the oscillometric method while participants were in the supine position after five minutes of rest. Two measurements taken three minutes apart were averaged and recorded. The results were reviewed by a specialist in medical chemistry who headed the protein–lipid laboratory at the University Medical Centre in Ljubljana [59].

### 2.4. Outcomes

In this secondary analysis, we aimed to assess the correlation between the intake of various plant-based food groups (per 50 g) and macronutrients (per 10 g) and selected CV risk factors among 151 individuals following a WFPB diet.

Various food groups were considered to assess their correlation with CV risk factors. These food groups included vegetables (processed and unprocessed separately), fruits (processed and unprocessed separately), whole grains (and products), legumes, potatoes, nuts/seeds, PB MRs, bread and bakery products, spices/herbs, pasta (made from white flour), fast food (ready-to-eat meals, processed vegetables, sugary foods, sugary drinks, vegetable-based fats, and sweeteners), sweet products, alcoholic drinks, vegetable fats, and sweeteners (all of which showed statistical significance) [59,60]. However, only food groups that showed a characteristic correlation are highlighted in the correlation table for food groups, macronutrients, and CV risk factors.

The macronutrients that were included in the correlation assessment with CV risk factors were carbohydrates, fiber, proteins, total fats, SFAs, MUFAs, and PUFAs. Furthermore, to better understand dietary habits, we have presented the energy content and nutritional composition of the two most frequently consumed foods from each of the nine food groups. Finally, we evaluated the correlation of different food groups and macronutrients with CV factors, including total cholesterol, LDL-C, HDL-C, non-HDL-C, triglycerides, systolic and diastolic blood pressure, and uric acid.

## 3. Statistical Analysis

Statistical analysis was performed using R 4.1.1. with the tidyverse [72], ggstatsplot [73], and arsenal [74] packages. The data were normalized and examined for approximate normality, and we carried out linear regression with multivariable adjustments (including age, sex, current BMI, smoking status, PA, and years on a plant-based diet). The significance threshold was set at *p* < 0.05, and no missing data were present. We did not perform a sensitivity analysis. The data are presented as the means (standard deviations).

## 4. Results

On average, the study participants showed a high consumption of vegetables (455 g/d), legumes (166 g/d), (nonfried) potatoes (140 g/d), nuts/seeds (52 g/d), and spices/herbs (32 g/d). On average, they preferred cooking, stewing, and roasting on a baking sheet as their usual food preparation methods. Vegetable oil consumption was infrequent and mostly restricted to salad preparation. Additionally, the average daily intake of vegetable oils was only one gram. The participants mostly used cooking, stewing, and roasting potatoes on a baking sheet as food preparation methods [59]. The average sodium intake was 2043 g/day, with the primary source being iodized salt. The participants’ average daily vitamin C intake was 337 mg/day, potassium intake was 4933 g/day, magnesium intake was 895 mg/day, and calcium intake was 1081 mg/day. The average intake of plant-based food groups is presented in Table 2.

The dietary pattern used (Table 3) resulted in a high intake of fiber (approximately 70 g per day) and a low intake of SFAs (approximately 7.5 g per day or 3% of energy intake). Previous publications contain comprehensive and detailed information about food groups and dietary intake categorized by sex. The results are standardized to 2000 kcal/d [59,60].

On average, the study participants displayed a standard blood lipid profile and blood pressure levels (Table 4) and had a normal BMI, with the majority being nonsmokers or former smokers (Table 5) and showing low stress levels (PSQ score, 0.29) and good sleep quality (PSQI score, 2.7). Additionally, during the study, they engaged, on average, in a relatively high amount of PA, according to the recommendations (metabolic equivalent (MET), 5541 MET minutes/week on average vs. the recommended 3000 MET minutes/week from a combination of walking-equivalent, moderate-intensity PA or vigorous-intensity PA). Furthermore, the participants performed resistance training 2.7 times/week, with each session lasting for at least 30 min [61].

Table 6 outlines the nutritional content of selected nutrients in the two most commonly consumed foods (per 50 g) across each food group to highlight the nutritional diversity and quality of these foods.

Notably, all the significant associations we observed exhibited weak or very weak coefficients, with values ranging from 0.16 < r < 0.32. All adjusted variables are presented in Table 6. Notably, linear regression was adjusted for sex, current BMI, smoking status, PA, and years on a plant-based diet. Nevertheless, regarding the intake of food groups (all per 50 g), we found that whole grain and fruit intake were negatively associated with total C, LDL-C, and HDL-C levels. Moreover, legume and nut/seed intake were only negatively associated with HDL-C. Interestingly, spice/herb intake was negatively associated with uric acid, but plant-based fast food and pasta (made from white flour) intake were positively associated with uric acid. The plant-based fast food group comprises a variety of foods and drinks, including ready-to-eat meals, processed vegetables, sugary products, vegetable-based fats, sweeteners, sugary drinks, and alcoholic beverages. Interestingly, none of these food groups significantly impacted triglyceride levels.

Regarding the intake of macronutrients (all per 10 g), all micronutrients, including fiber, were negatively associated with total C and HDL-C, with no significant association with LDL-C, except for carbohydrates and PUFAs. Furthermore, the intake of all macronutrients, except for total fat intake, showed a positive association with uric acid levels, with SFAs having the most significant impact, followed by carbohydrates. No macronutrient intake had a significant effect on triglycerides. Statistically significant correlations between food groups, macronutrients, and CV risk factors are presented in Table 7.

## 5. Discussion

### 5.1. Main Findings

The present study provides a comprehensive understanding of the impact of various food groups and macronutrients on selected CV risk factors in individuals following a long-term WFPB lifestyle. We expected that individuals who follow the WFPB lifestyle program would have blood lipid, blood pressure, and BMI profiles that meet the recommendations in evidence-based guidelines. In addition, as part of this secondary associative evaluation analysis, we predicted that high dietary fiber intake and low SFA consumption would contribute to favorable CV risk factor outcomes.

Regarding our first hypothesis, this study partially confirmed that a well-designed plant-based diet rich in dietary fiber and low in SFAs is associated with favorable markers of CV health. Furthermore, these findings underscore that even small quantities of unhealthy food groups can negatively affect specific CV risk markers, regardless of the overall dietary pattern. The phrase “small quantities” is often used to describe the general intake of foods in a typical Western diet but does not have a clear definition. However, the results of a recent secondary analysis of a randomized clinical trial with 244 participants who were randomly assigned to either a well-designed WFPB diet or a control group for 16 weeks suggested that minimizing the consumption of animal foods and animal products, sweets, and vegetable oils could be an effective strategy for weight loss in overweight adults. However, it is difficult to precisely determine the amount of these foods that can have clinically relevant adverse health effects, as published data are limited. The researchers analyzed both groups together, which made it impossible to determine the impact that “small quantities” of these food groups had on both studied groups. It is a poorly defined term compared to the typical Western diet, which is estimated to consist of small quantities of these foods [77,78]. However, the clinical significance of these minor effects found in our study is likely negligible within the context of an overall well-designed nutritional plan.

Interestingly, we did not find some of the expected associations between specific food groups, macronutrients, and CVD risk factors. For example, anticipated links between legumes, nuts/seeds, fiber, SFAs, and LDL-C levels have not been confirmed. This might be attributed to the relatively homogenous nature of the study sample, which limits the diversity of the results. Notably, our study population has much smaller variation than studies on individuals following a typical Western diet/lifestyle, where the relationships between healthy and unhealthy versions of the plant-based diet (indexes), as well as varied omnivorous diets, can be investigated, as well as the impacts of the different ratios of plant and animal food sources on various aspects of health [47,48,49,50]. Furthermore, this may be due to the overall nutritional sufficiency (on average, very high fiber intake and very low SFA intake) associated with a well-designed plant-based diet and the effective control of associated CV risk factors. Nevertheless, these findings emphasize that the overall healthfulness of an individual’s dietary pattern can significantly influence how individual food groups impact CV risk factors, providing valuable knowledge in this area.

### 5.2. Food Groups Associated with CV Risk Factors

The relationships between plant-based food groups and macronutrients and various CV risk factors are complex and multifaceted, mainly because they accumulate the effects of what we eat, what we do not eat, and how we live [79,80]. Nonetheless, while we found some significant associations, it is essential to emphasize that the observed relationships were generally weak. Specifically, we observed that a greater intake of whole grains and fruits was associated with lower total C, LDL-C, and HDL-C levels, indicating a potential protective effect against adverse blood lipid profiles; these findings are consistent with the findings of meta-analyses of these two food groups. Researchers have consistently identified their role in preventing CVDs [41,42,81,82].

In addition, legumes and nuts/seeds showed some unexpected negative associations, albeit only with HDL-C. While HDL-C has traditionally been regarded as an antiatherosclerotic biomarker, levels lower than the recommended levels of HDL-C represent a risk factor for CVDs [83,84]. However, recent studies have proposed a more complex mechanism—i.e., U-shaped—association between HDL-C and CV risk [83,85,86]. Based on findings from a pooled analysis of 37 prospective cohort studies, it is essential to pay attention not only to individuals with low HDL-C levels but also to those with relatively high HDL-C levels since both subpopulations have an increased risk of mortality [83]. Nevertheless, our participants had relatively high HDL-C levels (i.e., mean 1.4 mmol/L), and the recommended non-HDL-C level (2.6 mmol/L) still falls within the optimal range, which is associated with the lowest mortality risk [83,86,87]. This finding can also be attributed to a physically active lifestyle (5541 MET minutes/week) [61] and weight loss (an average of 7.1 kg) compared to the pre-WFPB lifestyle period. Individuals with a normal BMI lost 2.5 kg on average, overweight individuals lost 7.2 kg, and obese individuals lost 16.1 kg [88].

Regardless, the effect sizes of these dietary factors on CV risk factors were relatively modest. This is particularly notable given that the average daily intake of legumes (including soy foods) was 166 g (corresponding to 1162 g/week), and nuts/seeds were consumed at an average of 52 g (the main sources of nuts and seeds were walnuts, hazelnuts, flax, and sesame seeds) [59]. The relationship between CV risk factors and food group intake was assessed for every 50 g of these foods consumed. Additional legume intakes based on already very high levels (i.e., 1162 g/week) are supposedly above the threshold for additional clinical benefits compared to the scenario of moving from a legume intake being too low towards the recommended intake. Beans, lentils, chickpeas, and soybeans (tofu) were the most commonly consumed subgroups in the legume group [59]. In support of these findings, a recent systematic review and dose–response meta-analysis attempted to identify the optimal legume intake level for reducing CVD risk. The study estimated that 400 g per week provides the optimal CV benefit, beyond which the benefits appear to level off [19]. In addition, a review of 26 randomized controlled trials revealed that consuming pulses can significantly lower LDL-C levels compared to consuming controlled diets. The review estimated that a median intake of 130 g/day (i.e., 910 g/week) of pulses could lead to a 5% reduction in LDL-C, and a 5–10% reduction is expected from heart-healthy diets alone. It is worth noting that the authors highlighted that most trials included in the review were conducted on a foundation of heart-healthy diets. These diets consist of more than 20–25 g/day of fiber (2.8–3.5 times less than in our study) and less than 10% of the energy from saturated fat (up to 3.3 times more than in our study) [89]. Therefore, the cumulative intake of legumes needs to be considered in terms of its overall effect on HDL-C and LDL-C levels and other aspects of health.

Interestingly, we found that spice/herb intake was negatively associated with uric acid levels, potentially affecting CV health. The participants recorded their intake of all locally grown spices, herbs, bulbs, celery, rosemary, and turmeric [60]. Conversely, the intake of pasta (made from white flour) and plant-based fast food (usually a source of added oils/fat and free/added sugar) was positively associated with increased uric acid levels, suggesting that these food choices could contribute to increased CV risk through mechanisms related to uric acid metabolism [90,91]. While the observed effect size was statistically significant, it is essential to note that the effect size was relatively small. Furthermore, our study participants had limited consumption of these two food groups on average, with an average daily intake of only 17 g of white flour pasta and 6 g of PB fast food.

Importantly, our study participants had relatively high average potato consumption, averaging 140 g daily. However, despite this, we did not observe the typical adverse association with uric acid levels found in a meta-analysis of population-based cohorts [92]. The likely reason for this discrepancy lies in the method of preparation. In the group following the WFPB lifestyle program, the recommended methods of potato preparation were healthy and included cooking, steaming, or baking on baking paper—without frying or adding extra fat (i.e., vegetable oils or butter).

Our study did not reveal a significant relationship between food groups and systolic blood pressure. This could be attributed not only to the plant-based diet per se but also to its well-designed nature (predominantly from whole-food sources) and the influence of lifestyle factors, such as BMI, PA, nonsmoking status, or smoking cessation [93,94,95]. We considered factors, such as appropriate sodium and high potassium intake (averaging 2043 mg/day and 4933 g/day, respectively) and regular consumption of herbal teas [59], which likely played a role in these findings, in addition to a high intake of many micronutrients, which are also associated with controlled blood pressure (for example, vitamin C (in too low intake), magnesium, and calcium) [96]. Several studies conducted on the general population have shown that dietary supplements containing vitamin C, magnesium, and calcium favorably impact blood pressure control [97]. In our previous publication, we presented the results of a detailed assessment of nutrient intake separately from foods, dietary supplements, and PB MRs. Specifically, we standardized the estimated dietary intake to 2000 kcal/day; therefore, individuals’ average vitamin C intake from food, dietary supplements, and PB MRs was 197 mg/day and 152 mg/day, respectively. Similarly, magnesium intake was 616 mg/day and 272 mg/day, and calcium intake was 690 mg/day and 408 mg/day, respectively [60]. Hence, the well-controlled blood pressure in our study participants (115/71 mmHg) may be partly due to the WFPB lifestyle, especially dietary intake (whole foods consumed and highly processed food omitted), as well as PA, sleeping, nonsmoking, and stress management.

The final relationship we observed between food groups and CV risk factors was a positive association between PB MR and diastolic blood pressure. However, the association was weak, and the effect size was relatively small (i.e., 1 mmHg) based on an intake of 50 g per day. Our study participants consumed approximately 43 g of this specific food group per day.

Notably, recently, the researchers performed a secondary analysis on a clinical trial involving overweight adults randomly assigned to either a vegan or omnivorous diet. The study used plant-based diet indices and 3-day dietary records. Vegetable oils were inversely associated with body mass loss within the study sample [77]. However, further studies need to clarify how and at what quantity of intake potentially unhealthy (ultra-processed/refined) plant-based food groups (e.g., fruit juice, sugar-sweetened beverages, sweets, refined grains, or French fries) impact body mass (change) and CV risk factors within the vegan group, as well as their clinical relevance.

### 5.3. Macronutrients Associated with CV Risk Factors

Our observations revealed that a greater intake of fiber, which is a crucial component of plant-based diets and is derived from whole-food grains, legumes, fruits, vegetables, and nuts/seeds, with an average intake of 70 g per day, was associated with lower levels of total C and HDL-C. However, only the intake of PUFAs was significantly associated with lower LDL-C levels. A systematic review with a meta-analysis of 38 studies indicated a positive association between total C intake and total blood cholesterol levels. Notably, the most significant reduction in risk for various relevant outcomes was observed when daily dietary fiber intake was within 25 to 29 g [98]. However, our study revealed a notably high average fiber intake. This could explain why we did not find a typical beneficial association between fiber intake (per 10 g) and LDL-C levels. In a recent study, we examined 80 health-conscious adults who consumed either a plant-based or nonplant-based diet. We examined the relationship between the intake of fiber (75 g/day vs. 34 g/day) or SFAs (3% of energy intake vs. 9% of energy intake) and LDL-C levels (1.7 mmol/L vs. 2.8 mmol/L). This information was briefly mentioned in the introduction. Through multivariate linear regression adjusted for age, sex, and smoking status, the analysis revealed that fiber and SFA intake alone accounted for 43% of the variation in LDL-C levels, and age contributed an additional 4%. Notably, each 10 g increase in total fiber intake was associated with a 0.12 mmol/L reduction in LDL-C levels [51]. Furthermore, our study revealed a noteworthy relationship between PUFAs and lower LDL-C levels. The beneficial effect of PUFAs on LDL-C is well known [99,100]. However, it should be emphasized that PUFAs also include eicosapentaenoic acid/docosahexaenoic acid (average intake of 565 g per day) based on a greater intake of nuts/seeds, including flaxseed [59], which also has a beneficial effect [101]. Notably, a relatively weak positive correlation was detected between carbohydrate intake and LDL-C levels. This is probably because our participants consumed the majority of carbohydrates from whole-food sources. Additionally, the typical source of PUFAs (which also showed beneficial effects on LDL-C) was nuts and seeds (52 g, including walnuts and flaxseeds) and legumes (66 g, including soy), which also represent a typical source of carbohydrates. Therefore, this combination had a favorable effect on the negative correlation between the intake of carbohydrates and LDL-C.

Consistent with the expected associations between specific food groups and HDL-C levels, we also observed negative associations between all macronutrients and HDL-C levels (discussed above in the food groups section). We revealed an intriguing but weak negative correlation between SFA consumption and LDL-C levels. However, two key factors likely influenced the observed correlation. The study participants had a notably low average SFA intake—just 7.5 g/day or 3% of their total energy intake. Additionally, the subjects exhibited relatively low LDL-C levels, averaging 2.0 mmol/L. A prospective urban–rural epidemiology (*PURE*) observational study revealed a significant association between a higher intake of saturated fat and a reduced risk of stroke, which could explain our results. Although this contradicts the general belief that SFA increases LDL-C levels, the PURE study included subjects from different countries with varying socioeconomic backgrounds. For instance, the study included approximately 80,000 Asian and 50,000 non-Asian participants. In predominantly Asian individuals, SFAs constitute only 2.3–3.9% of the total energy consumed and are associated with general malnutrition [102]. In a recent study, researchers evaluated the dietary intake of a well-designed theoretical WFPB diet. The results of their research on participants consuming a 2000 kcal diet per day hardly achieved 3% of their energy from SFAs [53], which further explains the associations observed in our study and the PURE study. In addition, in our previous study mentioned above [51], we found strong correlations between lower/higher intakes of SFAs, fiber intake, and LDL-C levels. We explored the differences in LDL-C between individuals on a plant-based diet (consuming only 3% of their energy from SFAs) and individuals on an omnivorous diet (consuming approximately 9% of their energy from SFAs) via multivariate regression analysis [51].

Furthermore, regarding macronutrient intake, we observed positive associations with uric acid levels for all macronutrients except for total fat, MUFAs, and PUFAs. Notably, SFAs had the most pronounced negative impact on uric acid levels, followed by carbohydrates. Caution is crucial when interpreting the potential effects of the amount of SFAs consumed on uric acid levels. This is because, out of the sample size of 151 individuals, 72% were women. Women generally exhibit lower uric acid values than men and have different reference ranges. Therefore, it is essential to consider this when analyzing the results. As a result, the data provided in our case can only be utilized to estimate the effect of SFAs, and it is important again to emphasize their very low intake in our participants (3% of energy). However, further research is needed to determine the potential clinical relevance of these findings.

Our findings highlight the vital importance of maintaining a healthy diet to decrease cardiovascular (CV) risk factors. In a recent umbrella review of nine published meta-analyses, which included both observational and randomized controlled studies, researchers explored the impact of plant-based diets (vegetarian and vegan) on cardiovascular health. The study’s findings revealed a clear association between healthier dietary choices and a lower risk of cerebrovascular disease, CVD incidence, ischemic heart disease mortality, and ischemic stroke [103].

### 5.4. Strengths and Limitations

The strengths of our study are that it provides a comprehensive and detailed analysis of participants’ dietary intake, particularly their adherence to a well-designed plant-based diet, including the consumption of specific food groups and macronutrients. In addition, it includes a thorough assessment of participants’ CV health profiles, BMI, and lifestyle status. Such an approach allows a more holistic understanding of their health status. Furthermore, the participants represented a relatively homogeneous group in terms of diet and lifestyle, reducing the confounding variables and increasing the internal validity of our findings. For analyses, we adjusted for critical confounding variables, such as sex, BMI, smoking status, PA, and years on a plant-based diet, strengthening the credibility of the results.

However, this study’s cross-sectional design limits its ability to establish causation, as it can only show simultaneous associations between different variables measured. Furthermore, the limited sample size could constrain the generalization of the findings obtained to larger populations on a plant-based diet. It is worth noting that some of the data, such as dietary intake and PA, might be susceptible to recall and social desirability bias, as participants enrolled in this study were part of the WFPB lifestyle program. However, dietary intake was evaluated using a 3-day weighted dietary record in the most extensive sample of adults on a long-term WFPB lifestyle. The web-based software tool OPEN, designed for assessing the nutritional intake of recipes, was developed by the Jozef Stefan Institute, a renowned research institution. Additionally, there is a risk of inaccurate reporting due to under- or overreporting [104] and accuracy issues when entering menus into the system. In addition, this study focused on cholesterol and blood pressure as CV risk factors. Including a broader range of CV risk factors (e.g., LDL-C particles, fasting glucose, hemoglobin A1C, predicted insulin sensitivity index, oral glucose insulin sensitivity) and other health outcomes would provide a more comprehensive view of the impact of a diet practiced by the study participants.

## 6. Conclusions

Our study provides comprehensive insight into individuals who adhere to a WFPB lifestyle, namely, high vegetable, fruit, legume, potato, nut/seed, or spice/herb consumption and PA, while avoiding smoking (with the majority being nonsmokers or former smokers) and alcohol consumption. This results in a substantial fiber intake, low SFA consumption, and an active lifestyle. These dietary and lifestyle choices corresponded with favorable CV health profiles, encompassing, on average, normal lipid levels, blood pressure, and BMI. Additionally, the study participants engaged in a relatively high amount of PA compared to the general sedentary or physically nonactive population but within the recommended guidelines.

However, while we identified some significant associations between various food groups and CV risk/health factors, these were relatively weak with small effect sizes. These findings, adjusted for important confounders, suggest that CV health outcomes are influenced by a combination of dietary and lifestyle factors, underscoring the complexity of these interactions. Furthermore, they highlighted the need for careful dietary planning to optimize CV well being. Regardless of food group or macronutrient intake, our results emphasize that the composition and quality of a plant-based diet play vital roles in CV health, underscoring the multifaceted nature of these diet–health relationships.

## Figures and Tables

**Table 1 nutrients-16-00796-t001:** Basic plant-based dietary patterns [1,11,12,13,14,15].

	Plant-Based Diets	Vegan Diet	WFPB Diet	WFPB Lifestyle ^‡^
Does not include	Large amounts of animal food (the amount is not specified)	Flesh and animal food sources (meat, dairy, eggs, and fish). On average, a WFPB diet typically primarily excludes processed foods.		
Includes	A smaller amount of animal products compared to general dietary recommendations (the amount is not specified)	Plant and nonplant food sources ^†^	Whole-food, plant-based food sources ^††^	Whole-food, plant-based food sources ^††^
Description	This term usually includes vegetarian and vegan diets and should be used in combination with a comprehensive dietary explanation	It may contain processed foods high in caloric content, free sugars, fats, salt, preservatives, and/or refined flour. It may not necessarily incorporate whole foods. It can also include less appropriate food preparation methods.	This is a whole-food, plant-based diet that can be implemented with a low or high proportion of energy from fat-source foods.	It also includes a lifestyle, usually healthy and active
Dietary supplements	Depends on the quality of the planning and how restrictive the diet is regarding the elimination of animal food sources	At least vitamin B12, possibly eicosapentaenoic acid (EPA), docosahexaenoic acid (DHA), and vitamin D (depending on geographical areas with either lower or higher UV indices). Depending on the quality of the diet, there is the potential for deficient iodine and calcium intake ^†††^.		

WFPB diet = whole-food, plant-based diet, ^†^ Grains, legumes, vegetables, fruits, nuts/seeds, spices/herbs, fungi/mushrooms, some algae, bacteria. ^††^ The consensus regarding the percentage of energy derived from non-whole-food, plant-based diet sources is at the discretion of the authors and the individual’s interpretation of the strictness while still being able to classify the dietary pattern as a whole-food, plant-based diet. ^†††^ Other supplements, such as meal replacements (MRs), protein shakes, sports drinks, and other nutritional supplements, are not essential. However, people usually add more or less of them to any kind of diet. ^‡^ Studied an ongoing whole-food, plant-based lifestyle program.

**Table 2 nutrients-16-00796-t002:** Average intake of plant-based food groups.

Food Groups	Vegetables	Fruit	Whole Grains	Legumes	Potatoes	Nuts/ Seed	PB MRs	Spices/ Herbs	Pasta	PB Fast Foods
g/d	455 ± 190	363 ± 187	178 ± 114	166 ± 115	140 ± 123	52 ± 46	43 ± 72	32 ± 40	17 ± 35	6 ± 34

Vegetables: cruciferous, leafy green, and colored vegetables; fruits: unprocessed, berries included; grains: whole grains; legumes: beans, lentils, and soy; nuts/seed: walnuts, sesame, and flaxseed; spices/herbs: fresh and dried; pasta: from white flour; PB MRs = plant-based meal replacements; PB fast foods = plant-based fast foods: ready-to-eat meals, processed vegetables, sugary products, sugary drinks and alcoholic beverages, vegetable-based fats, and sweeteners.

**Table 3 nutrients-16-00796-t003:** Average energy and macronutrient intake.

Energy Intake, Macronutrients	Energy	Fiber	Carbohydrates	Protein	Total Fat	SFA	MUFA	PUFA
kcal/d, g/d, mg/d	2057 ± 689	70 ± 21	295 ± 101	78 ± 25	47 ± 23	7.5 ± 3.6	13 ± 8	20 ± 11
% E		7 ± 1	57 ± 5	15 ± 2	20 ± 5	3 ± 1	6 ± 2	9 ± 3

Kcal/d = kilocalories per day, g/d = grams per day, mg/d = milligrams per day, % E = percent per daily energy intake, SFAs = saturated fatty acids, MUFAs = monounsaturated fatty acids, PUFAs = polyunsaturated fatty acids.

**Table 4 nutrients-16-00796-t004:** Cardiovascular risk factor status.

CV Risk Factors	Total C	LDL-C	HDL-C	Non-HDL-C	Triglycerides	Systolic BP	Diastolic BP	Uric Acid
mmol/L, mmHg, µmol/L	3.7 ± 0.8	2.0 ± 0.7	1.4 ± 0.4	2.6 ± 0.5	0.9 ± 0.4	115 ± 11	71 ± 9	273 ± 68

CV = cardiovascular, total C = total cholesterol, LDL-C = low-density lipoprotein cholesterol, HDL-C = high-density lipoprotein cholesterol, BP = blood pressure.

**Table 5 nutrients-16-00796-t005:** Adjusted variables.

Adjusted Variables	Mean Age (Years)	Sex n (F/M)	Current BMI (kg/m^2^)	Smoking % (Never)	PA (Total METs)	WFPB Lifestyle (Mean Years)
	39 ± 13	109/42	23.9 ± 3.8	78	5541 ± 4677	4.1 ± 2.5

BMI = body mass index; PA = physical activity; WFPB = whole-food, plant-based.

**Table 6 nutrients-16-00796-t006:** The energy content and nutritional composition of selected nutrients (per 50 g) for the most commonly consumed foods within each of the nine food groups [66].

	Vegetables		Fruits		Whole Grains		Legumes		Potatoes	Nuts/Seeds		PB MR	Spices/Herbs		Pasta
Food group (per 50 g)	Broccoli/tomato		Berries/apples		Oatmeal/bread ^wg^		Beans/tofu		White potatoes	Walnuts/flaxseeds		PB MR	Onions/parsley		Pasta
Energy (Kcal)	14	10	32	30	197	120	165	81	36	360	245	199	27	46	163
Macronutrients															
Carbohydrates (g)	1.3	2.0	7.3	7.2	35	23	28	0.8	7.4	5.5	14	16	5.7	3.7	33
Dietary fiber (g)	1.5	0.2	1.2	1.0	6.8	1.0	15	0.2	1.0	3.1	19	9	0.9	2.1	1.5
Total fat (g)	0.1	0	0.2	0.1	3.8	0.8	0.5	5.1	0.1	35	15	4.8	0	0.2	0.8
SFA (g)	0	0	0	0.1	0.6	0.4	0	0.7	0.01	3.4	1.5	0.9	0.1	0	0.1
MUFA (g)	0	0	0	0	1.4	0.4	0	0	0	5.8	2.8	1.0	0.1	0	0.1
PUFA (g)	0	0.1	0.1	0.1	1.3	0.7	0	0	0	21	10	2.8	0	0.1	0.3
LA (g)	0	0	0	0	1.1	0.4	0	0.2	0	17	2.9	0	0	0.1	0.3
ALA (g)	0	0	0	0	0	0.1	0	0.1	0	4.5	11	0	0	0	0
Protein	1.9	0.4	0.4	0.1	5.8	4.8	12	8	1.4	7	12	17	0.8	7.2	5.8
Micronutrients															
*Vitamins*															
B9 (folate) (µg)	57	11	3.0	3.8	44	15	0	0	11	38	43	125	5.5	74	21
C (ascorbic acid) (mg)	24	12	5	2	0	0	0	0	10	1.3	0.3	60	9.8	19	0
Retinol equ. ^re^ (mg)	0.1	0.1	0	0	0	0	0	0	0.4	0	0	0.6	0	0.9	0
E (α-tocopherol) (mg)	0.3	0.4	0.3	0.3	0.7	0.4	0	0	0	0.9	0.2	0.9	0	1.8	0.1
*Minerals*															
Calcium (mg)	29	4.8	3.0	12	19	21	94	90	12	43	99	230	13	89	10
Magnesium (mg)	9.0	5.5	3.0	18	65	30	70	30	55	64	196	215	4.8	22	26
Phosphorus (mg)	31	12	6.0	5.5	151	75	298	0	25	204	331	0	21	43	94
Potassium (mg)	128	104	38	60	168	765	765	0	208	272	362	785	89	658	111
Sodium (mg)	11	1.7	0.5	0.6	3.4	0	12	1	1	1.2	30	0	1.3	86	1
*Trace elements*															
Iron (mg)	0.4	0.3	0.1	0.3	1.4	1.9	4.7	0	0.9	1.2	4.1	11	0.5	1.8	0.6
Iodine (µg)	7.5	0.6	0.5	0.4	2.3	3.0	0	0	1.2	1.5	5.0	115	0.9	1.7	0
Zinc (mg)	0.2	0.1	0.1	0.1	1.2	0.7	1.3	0	0.7	1.3	2.8	6	0.1	0.4	0.7
Selenium (mg)	0.4	1.0	0.1	0.7	4.8	1.2	1.6	0	0	2.7	13	39	0	0.5	32

^re^ Retinol equivalents = vitamin A + α-carotene (1 mg retinol equivalent = 12 mg α-carotene) + β-carotene (1 mg retinol equivalent = 6 mg β-carotene) + γ-carotene (1 mg retinol equivalent = 12 mg γ-carotene) [75]. ^wg^ = whole-grain bread, PB MR = plant-based meal replacement [76].

**Table 7 nutrients-16-00796-t007:** Correlations between food groups and macronutrients and CV risk factors.

	**Total C** **(mmol/L)**		**LDL-C** **(mmol/L)**		**HDL-C** **(mmol/L)**		**Systolic BP** **(mmHg)**		**Diastolic BP** **(mmHg)**		**Uric Acid** **(µmol/L)**	
Food groups (per 50 g)	↑↓	*p*-value	↑↓	*p*-value	↑↓	*p*-value	↑↓	*p*-value	↑↓	*p*-value	↑↓	*p*-value
Whole grains	−0.10	0.001	−0.05	0.032	−0.03	0.017						
Fruits	−0.22	0.024	−0.23	0.003	−0.02	0.004						
Legumes					−0.03	0.033						
Nuts/seeds					−0.10	0.003						
Spices/herbs											−3.6	0.013
PB MR									1.0	0.044		
PB fast food											3.9	0.017
Pasta											3.9	0.023
Macronutrient (per 10 g)	↑↓	*p*-value	↑↓	*p*-value	↑↓	*p*-value	↑↓	*p*-value	↑↓	*p*-value	↑↓	*p*-value
Fiber	−0.12	0.001			−0.06	0.001					7.9	0.006
Carbohydrates	−0.03	0.001	−0.01	0.021	−0.01	0.001					18	0.003
Protein	−0.09	0.001			−0.04	0.004					5.1	0.038
Total fat	−0.10	0.001			−0.05	0.001						
SFAs	−0.68	0.001			−0.31	0.001					38	0.019
MUFAs	−0.26	0.001			−0.12	0.001						
PUFAs	−0.25	0.002	−0.10	0.037	−0.12	0.002						

↑↓ = the direction of association (an arrow pointing up means a positive association, while an arrow pointing down means a negative association). LDL-C = low-density lipoprotein cholesterol; HDL-C = high-density lipoprotein cholesterol; PB MR = plant-based meal replacement; PB fast food = plant-based fast food; SFAs = saturated fatty acids; MUFAs = monounsaturated fatty acids; PUFAs = polyunsaturated fatty acids. The data were adjusted for age, sex, current BMI, smoking status, PA, and years on a plant-based diet.

## Data Availability

The data used to support the findings of this study are included within the article.

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
