# Peer review of "Associations of Dietary Intake with Cardiovascular Risk in Long-Term “Plant-Based Eaters”: A Secondary Analysis of a Cross-Sectional Study"

_nutrients, 2024, doi:10.3390/nu16060796_

Round 1

Reviewer 1 Report

Comments and Suggestions for Authors

Dear authors,

I have carefully studied the manuscript entitled “Associations of Dietary Intake with Cardiovascular Risk in 2 Long-term “Plant-based Eaters”” by Boštjan Jakše et al. The topic is very interesting, but there are some adjustments that need to be made:

1.     Line 20 – in “this” secondary analysis? It has not been mentioned anything before about this.

2.     Line 65-66 – I believe this should be written in the past tense.

3.     Line 84—86 – please do not use “;” for enumeration. Instead, use “,”.

4.     Line 102-107 – The authors expectations should rather be written in the discussion or conclusion part, rather than in the introduction.

5.     Line 111-114 – please give all the information needed in this article regarding the materials and methods. Each submitted manuscript should include all these details. Please include all the details regarding the questionnaires that were used, how did the 3-day dietary records were evaluated, what lipid profile parameters were included and what were the normal ranges, how was the blood pressure measured and quantified.

6.     Please write more details regarding all the cardiovascular risk factors of the patients included in this study.

7.     Please write a table that includes many types of plant-based food groups (per 50 g) and macronutrients (per 10 g). Readers do not know what type of food was evaluated.

8.     Line 168 – how did the authors counted a high consumption of vegetables/legumes/potatoes/nuts-seeds and spices/herbs.

9.     Line 174 – please detail the recommendations regarding the physical activity.

10.  Please write modalities of implementation of the plant-based diet in patients in patients with different cardiovascular disease (such as high blood pressure, heart failure, aortic stenosis).

11.  Can you please write more cardiovascular risk factors that can be modified after implementation of the plant-based diet?

12.  Do you have any laboratory data of the patients before and after the plant based diet?

13.  Line 178 – please include the quantities of and types of whole grains, types of fruits included in your study.

14.  Line 180 – please include the quantities and types of legumes and nut/seeds intake

15.  Line 181 – please include the quantities and types of herbs included in your study and which of them correlated with uric acid.

16.   Line 182 – please include the quantities and types of pasta included in your study.

17.  Line 182-183 – please describe the term plant based fast-food, with types and calories, macronutrients and SFA, PUFA.

18.   Line 182-186 – please provide examples for all of the types of food mentioned in these lines.

19.  Please include a table with macronutrients and micronutrients of all the type of food mentioned in this manuscript.

20.  Please rewrite table 1. I would like it to be more clearer (the authors can make more tables). For example, one can be just with the food groups, one with macronutrients, etc. It is very hard to read and understand.

21.    Table 2 – Please explain ­¯ sign. Does this mean the type of correlation (negative/positive)? Why are there 2 p-values? Why do the authors do not have all of the p-values in the table?

22.  Please describe what does well-designed plant-based diet means.

23.  Line 207 – please provide an evidence of what does small quantities of unhealthy food groups means.

24.  Line 230-238 – Please revise or give more examples and details regarding the correlation between low HDL and CV protection. All of us learn through our experience that HDL is a protective factor and decreases the CV risk.

25.  Line 241 – please add information regarding the physical activity that was performed in the study group. Moreover, what was the decrease in weight of the study population?

26.  Line 276 – you mentioned the type of food (potatoes) preparation. Please include a table with all the methods of cooking and quantities evaluated in your study.

27.  Line 277 – please add new paragraph when you write about blood pressure.

28.  Line 284-285 – please describe more regarding the correlations between blood pressure and vitamin C and magnesium. Did the researchers have evaluated these micronutrients in the study population?

29.  Line 323 – please prove more information regarding the correlation between uric acid and SFA

30.  Line 348 – I did not read any detailed analysis of the participant’s dietary intake. Moreover, please add more details regarding the CV health profile and lifestyle status of the evaluated study group.

Comments on the Quality of English Language

Small English requirements are needed.

Reviewer 2 Report

Comments and Suggestions for Authors

Dear Authors,
Your manuscript is well written and it deals with a very interesting topic. However, I think thay dome aspects need to be revised:
- in the introduction section, I suggest you add a table summarizing the characteristics of the main plant based dietary patterns;
- other dietary patterns ( such as Mediterranean Dietand Nordic Diet) have been associated with CV risk reduction, I think you should at least mention it;
- in the table describing the population characteristics, when you cite macronutrients carbohydrate intake is missing;
- the study partecipants' family history for CV, personal history of CV events, plasma levels of Lp(a) should be added;
- it is  not specified if the study partecipants are taking drugs to lower cholesterol and/or glucose plasma values;
- I think that a 3 day nutritional recall is not sufficient to state that different lipid profile is due to different nutrient intake in the study partecipants. I suggest that lipid, glucose and uric acid plasma values and blood pressure before starting the plant based diet should be reported and compared to those evaluated while consuming a plant based diet;
- a more detailed description of food matrices would be advisable;
Best regards

Comments on the Quality of English Language

Minor English check needed

Round 2

Reviewer 1 Report

Comments and Suggestions for Authors

Dear authors,

Thank you for all your responses. Thus, I have some more issues that were not resolved:

1.     I think it would be necessary to add a table that includes all the many types of plant-based food groups (per 50 g) and macronutrients (per 10 g). Readers do not know what type of food was evaluated. Moreover, please add types of SFA, MUFAs and PUFAs with all of the details mentioned above. I don’t believe just an enumeration is enough.

2.     Even if all your subjects included in the study were healthy, I still believe that writing some modalities of implementation of the plant-based diet in patients with higher cardiovascular risk would increase your article quality and the number of citations.

3.     Please include a table with macronutrients and micronutrients of all the type of food mentioned in this manuscript. I don’t believe this would be an uninformative and disturbing information, due to the fact that there are different types of fruits and vegetables that have very different macronutrients and micronutrients. For example, you can not compare a banana with a kiwi.

4.     Regarding table 6. I have understood the positive and negative correlation between the two factors, but I don’t understand why you have 2 values. If you have a negative correlation, you should have a negative p value. Please explain. Thank you!

5.     Line 257 - How did you evaluated the stress level of the participants? More data about the questionnaire used would be useful.

Reviewer 2 Report

Comments and Suggestions for Authors

Dear Authors,

I think you have improved your manuscript and it is now suitable for publication.

Best regards,
